Evaluation of awareness, attitudes, and practices towards disaster preparedness among Saudi healthcare professionals: implications for enhancing emergency response and training

Thirunavukkarasu Ashokkumar ashokkumar@ju.edu.sa 1
Alsaidan Aseel Awad 1
Aldhafeeri Sayer T. 1
Abdel-Salam Doaa Mazen 2
Aldhafeeri Nasser T. 3
Alanazi Muzun Ali 4
Almughamisi Randa Mansour 4
Alkawkbei Diyam Khalid 4
Al-Zahrany Wijdan 5
1 Department of Family and Community Medicine, College of Medicine, Jouf University , Sakaka , Aljouf , Saudi Arabia
2 Department of Public Health and Community Medicine, Faculty of Medicine, Assiut University , Assiut , Egypt
3 Department of Public Health, Public Health Authority , Hafr Al Batin , Saudi Arabia
4 Medical Student, College of Medicine, Jouf University , Sakaka , Aljouf , Saudi Arabia
5 Department of Family Medicine, Ministry of Health , Sakaka , Aljouf , Saudi Arabia
Chen Yung-Sheng
Electronic publication date: 2025 Dec 15
Publication date: 2025
Volume: 13
Electronic Location ID: e20464
Received 2025 May 28; Accepted 2025 Nov 3
Copyright: ©2025 Thirunavukkarasu et al.
Copyright year: 2025
Copyright holder: Thirunavukkarasu et al.
License: This is an open access article distributed under the terms of the Creative Commons Attribution License, which permits unrestricted use, distribution, reproduction and adaptation in any medium and for any purpose provided that it is properly attributed. For attribution, the original author(s), title, publication source (PeerJ) and either DOI or URL of the article must be cited.
License URL: https://creativecommons.org/licenses/by/4.0/

Keywords: Disaster, Preparedness, Healthcare professionals, Knowledge, Training, Saudi Arabia

Funding: The Deanship of Graduate Studies and Scientific Research at Jouf University DGSSR-2025-01-01071 This work was funded by the Deanship of Graduate Studies and Scientific Research at Jouf University under grant No. (DGSSR-2025-01-01071). The funders had no role in study design, data collection and analysis, decision to publish, or preparation of the manuscript.

==============================
Background and Aim

Disaster management and preparedness by healthcare professionals (HCPs) are integral to safeguarding public health. The present study assessed the awareness, attitude, and practice of disaster preparedness among HCPs of the central Saudi Arabia region. Furthermore, the present study determined the predictors associated with these three domains.

Methods

The present study was conducted among 390 HCPs from central Saudi Arabia using a cross-sectional design. The authors used a standard and validated data collection tool to gather the required information. We conducted the Spearman correlation analysis to identify the correlation among these three domains. Finally, the authors applied a multivariate analysis to identify the factors associated with the low levels of awareness, attitude, and practice.

Results

Among the HCPs studied, we observed a sizable proportion had low knowledge (36.2%), attitude (45.9%), and practice (49.2%) towards disaster preparedness. The present study showed a significant positive correlation between awareness and attitude (rho = 0.649) and awareness and practice (rho = 0.565). Nursing professionals had significantly higher awareness (adjusted odds ratio (AOR) = 3.187, p = 0.007), attitude (AOR = 4.564, p = 0.001), and practice (AOR = 3.235, p = 0.007) scores. Furthermore, married healthcare professionals had significantly higher practice scores (AOR = 4.102, p = 0.020).

Conclusion

There is a need to plan, design, and develop targeted educational programs to improve the awareness, attitude, and practice of the HCPs in disaster preparedness. Targeted interventions are essential to enhance HCPs’ preparedness for diverse disaster scenarios.

Introduction

The public health emergency management (PHEM) is a key element in the health systems because disasters are common in most countries and come in the form of natural and man-made disasters (Brencic et al., 2017; Rose et al., 2017). Lately, rampant urbanization and global population growth have led to an increase in natural and artificial disasters or calamities (He et al., 2021; UNDRR, 2019). The impact of the disaster can be abrupt and limited; however, it is very often extensive and could persist for longer durations. In addition, these challenges may prompt the suffering community or group to thrive under their own limited supplies, and thus may necessitate aid from outside, which possibly will involve immediate authorities, or even national and international government bodies (Dagnino, Qian & Beatrici, 2024; EM-DAT, 2024; Zeng et al., 2025). Emergency events database categorizes disasters or calamities into natural, which includes geological (tsunamis, volcanic eruptions, and earthquakes), meteorological (cyclones and storms), hydrological (floods), climatological (wildfires and droughts), disasters, and catastrophes triggered by human-driven activities like industrial (biochemical spill), transport, and miscellaneous accidents, for instance, detonation (EM-DAT, 2024; UNDRR, 2017).

Recognizing an outbreak depends on the government agencies’ ability to identify a significant rise in the cases of a specific disease, more than the number ordinarily expected. Epidemic prevention, preparedness, and response rely on a high level of technical expertise to implement the associated plans (Li et al., 2022). Healthcare professionals (HCPs) must be familiar with emergency management procedures, possess the skills to effectively integrate stakeholders, and meet the stress demands of their profession. Studies have shown that healthcare professionals’ attitudes towards disaster planning strongly influence their likelihood of engaging in preparedness training and following institutional procedures, as well as the overall implementation of emergency management plans (Kennedy et al., 2020; Li et al., 2022).

Disaster preparedness among HCPs can be defined as all the measures taken to ensure that the health care institution, its workers, and patients are ready for a given disaster, including training, drills, supply of essentials, and equipment, among others (UNDRR, 2017). A recent study of these practices provides insight into the real-world application of the intended preparedness efforts. It also records the discrepancies between proposed policies and implementations, which are crucial in modifying public health approaches and responses to disasters where they are needed most (Calonge, Brown & Downey, 2020). Literature review on the knowledge, attitude, and practice of disaster preparedness among the HCPs revealed that knowledge deficits, attitude towards disasters, and disaster preparedness can be improved, and practical disaster preparedness measures need to be improved (Alhamaid et al., 2024; Mohamed, Abdel-Aziz & Elsehrawy, 2023; Tassew et al., 2022). While there is some appreciation of disaster preparedness in Saudi Arabia, important areas that still require public health attention include training in disaster preparedness and the HCPs’ attitude towards it, as well as the lack of consistent enforcement of disaster preparedness practices (AlDulijand et al., 2024; Alhamaid et al., 2024; Almukhlifi et al., 2024). This has made it clear that proper disaster preparedness measures are necessary and important, especially in Saudi Arabia, due to the geographical and socio-political structure of the Middle East. Furthermore, plenty of natural calamities in Saudi Arabia could be possibly owing to the diverse geographical landscapes, such as gigantic mountains or valleys, and coastline sections, whereas other areas are tremendously arid (Alamri, 2011). Unlike most clinical conditions, disasters can occur unexpectedly without giving early warning signs. Therefore, preparation and education beforehand would certainly aid in the proper handling of these grave conditions along with their aftereffects.

It can be concluded that the rationale behind conducting this research is based on the recommendations of the literature review, existing knowledge gaps, and regional needs, as follows. To begin with, the applicability of the given study to the field of public health cannot be overemphasized because it is vital to know the preparedness and the attitudes of HCPs in this context (Alrowili et al., 2025; Skaff et al., 2024). Secondly, despite the increasing global recognition of disaster preparedness, Saudi evidence remains fragmented. Thirdly, much of the available evidence is confined to single institutions, specific professions (such as nurses or emergency medical technicians), or narrow aspects of preparedness, leaving critical gaps in understanding broader predictors across multiple cadres of HCPs (Al-Shammari et al., 2024; Alrowili et al., 2025). This limits the ability to design targeted, evidence-based training and preparedness strategies. Moreover, the Gassim region represents a significant context due to its diverse healthcare infrastructure and inclusion of both Saudi and expatriate professionals. By evaluating the predictors of awareness, attitudes, and practices among different cadres within the Gassim context, our study will provide not only context-specific but also relevant insights that can be applied by policymakers beyond the region to develop specific training and preparedness plans. Finally, an ongoing evaluation of the HCPs’ preparedness may provide policymakers with sufficient information to implement emergency response and training programs for the HCPs (Alrowili et al., 2025; Bajow et al., 2024). So that they will be prepared to handle any disaster efficiently and reduce short- and long-term public health impacts. Therefore, we performed this study to determine the HCPs’ knowledge, attitude, and practice towards disaster preparedness. The authors also determined the predictors related to these three domains.

Materials & Methods

Study description

This research employed a cross-sectional study design, which was conducted from October 2024 to March 2025. The investigation took place in the Gassim Province in Saudi Arabia, where it surveyed HCPs at different facility levels, such as primary healthcare centers and secondary hospitals, together with tertiary referral centers. This province is in the central part of Saudi Arabia. This region is one of the 13 provinces of Saudi Arabia, situated in the country’s central region. According to the latest available data, it has a population of 1,336,179 with an area of 58,046 km2. It ranks as the seventh most populous region, with a population density of 23 people per km2. Socio-economically, Gassim is characterized by relatively low poverty rates compared to other regions of Saudi Arabia, which may influence disaster preparedness and health system capacity (Abbakar, 2023; Brinkhoff, 2022). The study included HCPs from different cadres, including physicians, nurses, pharmacists, and lab technicians. The survey excluded individuals who had been newly joined (<6 months), those unwilling to participate, and those on leave at the time of data collection. Furthermore, we excluded the HCPs working only in academic and administrative settings.

Sampling description

The required number of HCPs was determined using the online sample size calculator. To obtain the maximum number of participants for a valid conclusion, we have used 50% as the expected proportion. Other metrics used were a 95% confidence interval and a 5% margin of error. It was found that a minimum of 384 HCPs were required to get a valid conclusion. The present study employed the convenience sampling method to recruit participants. This sampling was chosen due to the practical constraints of accessing HCPs across diverse facilities, a strategy commonly applied in research. Even though the convenience sampling may have its own limitations, this is a commonly used tool in the cross-sectional survey among HCPs, especially when the target population is not easily accessible in a systematic manner. This approach is consistent with similar KAP studies conducted on HCPs in resource-limited or time-sensitive contexts, as they applied convenience sampling to maximize response rates and ensure feasibility (Al-Qerem et al., 2023; Salih et al., 2025). The researchers visited the facilities in person, introduced the study purpose, and invited eligible professionals to participate. The research team continued to recruit participants until the preferred sample population had reached its full quota of representatives from various healthcare facilities and professionals.

Ethical statement

The team conducted this research in accordance with the principles of the Declaration of Helsinki. The regional ethics committee in the Gassim region, Saudi Arabia, issued the ethical approval letter to conduct this survey (Approval No: 607/46/3231, Dated: 30.09.2024). Each HCP was given a brief explanation of the study objectives, assurances of confidentiality, and their voluntary right to participate or decline without any consequences. Furthermore, we collected anonymous data and presented the overall results in this manuscript, not the individual data.

Data collection steps

After obtaining the informed consent form, the agreed HCPs were requested to complete the survey (Google form) on the data collectors’ electronic devices. The validated data collection form consisted of four sections. In the first part, we inquired about HCPs’ background details such as age, gender, work experience, work setting, qualification, and prior training in disaster management. These Independent variables were selected based on existing literature and contextual relevance to disaster preparedness among HCPs. In the second (awareness domain), the HCPs’ familiarity with the core concepts of disaster management and the system at the local and national levels. This domain consisted of nine items, to which the participants responded as “yes”, “no”, or “not sure”. We awarded 1 point for the answer yes, and the remaining responses received 0 points. The third (attitude domain) assessed participants’ beliefs, motivation, and personal commitment toward disaster preparedness in healthcare settings. The final (practice domain) was designed to evaluate the actual behaviors and self-reported readiness of HCPs in applying disaster preparedness principles in their work settings. Both the attitude and practice sections had ten questions. The HCPs responded on a 5-point Likert scale ranging between strongly agree (5 points) and strongly disagree (1 point). To make interpretation comparable across domains, the raw scores were converted into percentages by dividing the participant’s score by the maximum possible score and multiplying by 100.

In the end, the cumulative scores were obtained during analysis, and we grouped each domain into low (<60%), medium (60–79%), and high (≥80%). This proportion was calculated based on the cumulative scores of each domain. These thresholds are based on Bloom’s taxonomy of educational objectives and have been consistently adopted in previous public health research (Alanazi et al., 2023; ALRuwaili et al., 2024; Bloom, 1956). These categorical variables were subsequently used in regression analysis as ordinal outcomes, with the ‘low’ category serving as the reference group. The research team designed the applied data collection instrument through group discussions (with the relevant experts) and existing open-source evidence (content validity), with modifications to ensure contextual and cultural appropriateness for Saudi HCPs. The authors performed further validity and reliability analysis during the pilot study, which was performed among 33 different HCPs from various health facilities. All the HCPs involved in the pilot study indicated that the questionnaire was easy, locally suitable, and could be completed in less than 10 min (face validity). Data from the pilot study using the scales for knowledge, attitude, and practice were entered into the SPSS program to measure the Cronbach’s alpha of the scales used. The Cronbach’s alpha values for awareness, attitude, and practice were 0.87, 0.79, and 0.83, respectively, which is an acceptable level (Tavakol & Dennick, 2011). Furthermore, exploratory factor analysis confirmed the expected three-factor structure of the instrument (awareness, attitude, and practice). The satisfactory item loadings (>0.40) and total variance explained of approximately 61% indicate good construct validity.

Data analysis

The data gathered from the Excel sheet was transferred to the Statistical Package for Social Sciences (SPSS, V.22) for further processing. The data were coded in SPSS as per the coding sheet, and the statistical team performed the analysis. The data of all three domains did not fulfill the normal distribution criteria determined by Shapiro–Wilk’s test. Therefore, we used Spearman’s correlation test to determine the correlation among the domains. Finally, the predictors related to these domains were determined using the binomial regression model. This model adjusted all independent variables in a single step (enter method) to find the adjusted odds ratio and 95% confidence intervals. Collinearity diagnostics were performed, and variance inflation factor (VIF) values were < 2, indicating no concerning multicollinearity. We also set a p-value of less than 0.05 as statistically significant.

Results

The research team approached 447 HCPs during the data collection period, of which 390 agreed to participate in the survey (response rate is 87.2%). Of the 390 HCPs, the majority were in the age group of 31 to 45 years, females (53.6%), married (48.2%), Saudi nationals (63.3%), and working in general hospitals (45.1%). Nearly 37% of the participants had a bachelor’s degree, and 58.7% of the HCPs participated in disaster management training (Table 1).

Table 1 Background characteristics of healthcare professionals (HCPs) from the Gassim region, central Saudi Arabia (n = 390).

Characteristics	Frequency	Percentage	
Age (years)			
30 or less	122	31.2	
31 to 45	166	42.5	
46 and above	102	26.3	
Sex			
Male	181	46.4	
Female	209	53.6	
Married status			
Single	152	39.0	
Married	188	48.2	
Widow / Divorcee	50	12.8	
Education level			
Diploma	43	11.0	
Bachelor	144	36.9	
Masters	113	29.0	
Doctorate or equivalent	90	23.1	
Nationality			
Saudi	247	63.3	
Expatriates	143	36.7	
Healthcare professionals (HCPs) category			
Physicians	81	20.8	
Pharmacists	85	21.8	
Lab technicians	78	20.0	
Nurses	74	19.0	
Others	72	18.5	
Work experience			
Less than 5 years	165	42.3	
5 to 10 years	136	34.9	
>10 years	89	22.8	
Work setting			
Primary health center	164	42.1	
General hospital	119	27.4	
Specialty hospital	107	27.4	
Participation in disaster management training			
No	161	41.3	
Yes	229	58.7	

A majority of participants demonstrated awareness of key aspects of disaster preparedness, with over 60% affirming familiarity with programs, emergency risks, referral contacts, and evacuation protocols (Table 2).

Table 2 Distribution of participants’ responses in the awareness domain (n = 390).

Responses are shown as frequencies and percentages.

Awareness	Yes	No	
	n	%	n	%	
I am familiar with initiatives focused on disaster preparedness	247	63.3	143	36.7	
I am aware of sources where I can access research or information related to disasters	245	62.8	145	37.2	
I am knowledgeable about the potential emergency risks in this country (such as natural disasters, embargoes, war, etc.)	249	63.8	141	36.2	
I am aware of the appropriate contact to reach out to in the event of a disaster emergency (such as the health department)	235	60.3	155	39.7	
I understand the organizational structures and responsibilities of local and national agencies involved in disaster preparedness.	253	64.9	137	35.1	
I have enough experience in handling any disaster/emergency	209	53.6	181	46.4	
Disaster medicine is inherently a systems-focused field involving various responding organizations	197	50.5	193	49.5	
I am very aware of evacuation protocols from a building in case of any disaster occurrence	242	62.8	145	37.2	
Overall, my knowledge of disaster management is very high	249	63.8	141	36.2	

Table S1 shows the participants’ responses regarding their attitude towards disaster preparedness. Nearly 41% of the participants were interested in educational classes on disaster medicine preparedness (strongly agree—21.3%, agree—19.5%). Furthermore, 43.6% of the respondents believed that disaster preparedness is important for healthcare professionals (strongly agree—21.0%, agree—22.6%). Thinking that the workplace adequately prioritizes disaster preparedness was reported by 43.7% of the participants (strongly agree—24.2%, agree—19.5%). In addition, 45.4% of the respondents (strongly agree—22.1%, agree—23.3%) indicated that they felt supported by the organization in terms of disaster preparedness initiatives.

Table S2 depicts the participants’ responses in the practice section. Nearly 44% of the participants feel comfortable while executing evacuation protocols in the event of a disaster (strongly agree—21.5%, agree—22.6%). Additionally, 46.9% of the respondents reported that they are ready to practice under a disaster, even though some basic medications may not be available (strongly agree—25.1%, agree—21.8%). Regularly reviewing and updating their knowledge of disaster preparedness measures was done by 45.1% of the participants (strongly agree—25.1%, agree—20.0%).

Figure 1 depicts that 41% of the participants had medium knowledge. Furthermore, low attitude and practice were shown among 45.9% and 49.2% of the participants, respectively.

Figure 1 Distribution of healthcare professionals (n = 390) across levels of awareness, attitude, and practice towards disaster preparedness.

Categories were classified as low (<60%), medium (60–79%), and high (≥ 80%).

Table 3 showed significant positive correlations between knowledge and attitude (rho = 0.649, p = 0.001), knowledge and practice (rho = 0.565, p = 0.001), and attitude and practice (rho = 0.708, p = 0.001).

Table 3 Spearman’s correlation analysis between awareness, attitude, and practice (n = 390).

Correlation coefficients (rho) and p-values are reported.

Variable	rho*/p-value	
Knowledge–Attitude	0.649 (0.001)	
Knowledge–Practice	0.565 (0.001)	
Attitude–Practice	0.708 (0.001)	
Notes.

* Spearman’s rho value.

Table 4 shows the predictors associated with the HCPs’ awareness. Significantly higher levels of awareness were observed among nurses (adjusted odds ratio (AOR) = 3.19, 95% confidence interval (CI) [1.36–7.44], p = 0.007), and those HCPs who attended training programs related to disaster preparedness (AOR = 3.14, 95% CI [1.95–4.86], p = 0.037). Lower levels of awareness were observed among participants with PhD/MD education (AOR = 0.46, 95% CI [0.23–0.94], p = 0.032) and those in the age group of 31 to 45 years (AOR = 0.43, 95% CI [0.20–1.07], p = 0.006).

Table 4 Binomial logistic regression analysis of factors associated with awareness of disaster preparedness among HCPs (n = 390).

Adjusted odds ratios (AOR) and 95% confidence intervals (CI) are reported.

Variables	Total	Awareness	
		Low
n = 141	Medium/
High
n = 249	Adjusted odds ratio (AOR) (95% Confidence interval (CI))	p-value	
Age						
30 or less	122	61	61	Ref		
31 to 45	166	58	108	0.43 (0.26–0.60)	0.006	
46 and above	102	22	80	0.46 (0.20–1.07)	0.070	
Gender						
Male	181	67	114	Ref		
Female	209	74	135	1.37 (0.77–2.45)	0.281	
Marital status						
Single	152	66	86	Ref		
Married	188	59	129	1.44 (0.40–5.16)	0.577	
Widow/Divorced	50	16	34	0.53 (0.38–1.72)	0.577	
Education						
Diploma	43	18	25	Ref		
Bachelor	144	66	78	0.67 (0.24–1.83)	0.435	
Masters	113	40	73	0.33 (0.16–0.66)	0.002	
PhD/MD/Saudi board	90	17	73	0.46 (0.23–0.94)	0.032	
Nationality						
Saudi	247	99	148	Ref		
Non-Saudi	143	42	101	0.69 (0.43–1.12)	0.137	
HCWs category						
Physicians	81	33	48	Ref		
Pharmacists	85	35	50	1.46 (0.71–2.99)	0.310	
Lab technicians	78	21	57	1.39 (0.69–2.83)	0.358	
Nurses	74	23	51	3.19 (1.36–7.44)	0.007	
Others	72	29	43	2.51 (1.07–5.88)	0.034	
Work experience						
Less than 5 Yrs	165	59	106	Ref		
5 to 10 years	136	59	77	1.26 (0.54–2.94)	0.596	
>10 years	89	23	66	0.93 (0.40–2.18)	0.874	
Work setting						
Primary health center	164	65	99	Ref		
General hospital	119	42	77	0.71 (0.40–1.26)	0.241	
Specialty hospital	107	34	73	0.72 (0.39–1.34)	0.300	
Training					
No	161	69	92	Ref		
Yes	229	72	157	3.41 (1.95–4.86)	0.037	
Notes.

Ref reference category

Table 5 presents the predictors associated with the HCPs’ attitude. Healthcare professionals with work experience from 5 to 10 years (AOR = 0.411, 95% CI [0.18–0.96], p = 0.040), those with experience > 10 years (AOR = 0.38, 95% CI [0.17–0.53], p = 0.001), and those working in general hospitals (AOR = 0.44, 95% CI [0.25–0.79], p = 0.006) had significantly lower attitude scores. Similar to the awareness, significantly higher levels of awareness were observed among nurses (AOR = 4.56, 95% CI [1.87–11.27], p = 0.001), and those HCPs who attended training programs related to disaster preparedness (AOR = 2.77, 95% CI [1.43–3.80], p = 0.013).

Table 5 Binomial logistic regression analysis of factors associated with attitude towards disaster preparedness among HCPs (n = 390).

AOR and 95% CI are reported.

Variables	Total	Attitude	
		Low
n = 179	Medium /
High
n = 211	AOR (95% CI)	p-value	
Age						
30 or less	122	68	54	Ref		
31 to 45	166	87	79	0.69 (0.58–0.93)	0.054	
46 and above	102	24	78	0.511 (0.23–1.12)	0.092	
Gender						
Male	181	89	92	Ref		
Female	209	90	119	1.30 (0.71–2.38)	0.391	
Marital status						
Single	152	74	78	Ref		
Married	188	83	105	1.553 (0.44–5.45)	0.491	
Widow/Divorced	50	22	28	0.59 (0.28–1.29)	0.190	
Education						
Diploma	43	20	23	Ref		
Bachelor	144	77	67	0.74 (0.26–2.09)	0.571	
Masters	113	54	59	0.35 (0.18–0.71)	0.004	
PhD/MD/ Saudi board	90	28	62	0.48 (0.24–0.95)	0.034	
Nationality						
Saudi	247	123	124	Ref		
Non-Saudi	143	56	87	0.64 (0.39–1.04)	0.073	
HCWs category						
Physicians	81	45	36	Ref		
Pharmacists	85	43	42	0.94 (0.45–1.98)	0.874	
Lab technicians	78	20	58	0.98 (0.47–2.04)	0.952	
Nurses	74	38	36	4.56 (1.87–11.27)	0.001	
Others	72	33	39	1.07 (0.45–2.51)	0.885	
Work experience						
Less than 5 Yrs	165	72	93	Ref		
5 to 10 years	136	90	46	0.41 (0.18–0.96)	0.040	
>10 years	89	17	72	0.38 (0.17–0.53)	0.001	
Work setting						
Primary health center	164	89	75	Ref		
General hospital	119	47	72	0.44 (0.25–0.79)	0.006	
Specialty hospital	107	43	64	0.73 (0.39–1.37)	0.322	
Training						
No	161	84	77	Ref		
Yes	229	95	134	2.77 (1.43–3.80)	0.013	
Notes.

Ref reference category

Table 6 presents that the participants in the age group 31–45 years (AOR = 0.49, 95% CI = 0.28–0.71, p = 0.024), those with higher levels of qualification, such as master, PhD/MD/Saudi board (AOR = 0.25, 95% CI [0.13–0.50], p = 0.001 & AOR = 0.41, 95% CI [0.26–0.67], p = 0.006) had significantly lower practice scores. The significantly higher levels of practice were noted in non-Saudi nationals (AOR = 2.07, 95% CI [1.47–3.69], p = 0.031), nurses (AOR = 3.24, 95% CI [1.38–7.59], p = 0.007), and those who had training (AOR = 4.35, AOR = 2.53–6.31, p = 0.001).

Table 6 Binomial logistic regression analysis of factors associated with practice of disaster preparedness among HCPs (n = 390).

AOR and 95% CI are reported.

Variables	Total	Practice	
		Low
n = 192	Medium/
High
n = 198	AOR (95% CI)	p-value	
Age (years)						
30 or less	122	71	51	Ref		
31 to 45	166	75	91	0.49 (0.28–0.71)	0.024	
46 and above	102	46	56	1.450 (0.68–3.09)	0.336	
Gender						
Male	181	103	78	Ref		
Female	209	89	120	1.05 (0.58–1.89)	0.874	
Marital status						
Single	152	79	73	Ref		
Married	188	89	99	4.10 (1.25–6.42)	0.020	
Widow/Divorced	50	24	26	0.91 (0.44–1.89)	0.801	
Education						
Diploma	43	21	22	Ref		
Bachelor	144	88	56	0.63 (0.23–1.77)	0.383	
Masters	113	57	56	0.25 (0.13–0.50)	0.001	
PhD/MD/Saudi board	90	26	64	0.41 (0.26–0.67)	0.006	
Nationality						
Saudi	247	136	111	Ref		
Non-Saudi	143	56	87	2.07 (1.47–3.69)	0.031	
HCWs category						
Physicians	81	51	30	Ref		
Pharmacists	85	49	36	0.61 (0.30–1.26)	0.181	
Lab technicians	78	22	56	0.76 (0.37–1.55)	0.450	
Nurses	74	36	38	3.24 (1.38–7.59)	0.007	
Others	72	34	38	0.93 (0.40–2.16)	0.874	
Work experience						
Less than 5	165	71	94	Ref		
5 to 10	136	82	54	1.08 (0.49–2.39)	0.849	
>10	89	39	50	0.53 (0.24–1.17)	0.113	
Work setting						
Primary health center	164	86	78	Ref		
General hospital	119	52	67	0.83 (0.47–1.46)	0.513	
Specialty hospital	107	54	53	1.07 (0.58–1.97)	0.849	
Training						
No	161	99	62	Ref		
Yes	229	93	136	4.35 (2.53–6.31)	0.001	
Notes.

Ref reference category

Discussion

HCPs are considered the front-line staff in normal situations. However, their responsibilities increase several-fold during disasters and emergencies at hospital sites. Saving human lives and promoting their health in emergencies requires high efficiency and appropriate skills associated with realistic capabilities of healthcare professionals (Khirekar et al., 2023; National Academies of Sciences, Engineering, and Medicine, National Academy of Medicine & Committee on the Future of Nursing 2020–2030, 2021; Olorunfemi & Adesunloye, 2024). Many types of disasters are liable to occur in the Kingdom of Saudi Arabia, including Middle East respiratory syndrome (MERS), landslides, earthquakes, storms, floods, and human-made disasters (annual mass of pilgrims’ aggregations, accidents related to the oil industry) (World Bank Group, 2021; Al-Bassam, Zaidi & Hussein, 2014; Apostolopoulos et al., 2024; Palacios, Palacios-Rosas & Abdul-Aziz-Al-Mughanam, 2024; Rahman et al., 2017). Understanding the knowledge, attitude, and practice of healthcare professionals regarding disaster preparedness is essential in guiding policymakers to develop strategic plans that enhance disaster preparedness.

The present study revealed that most healthcare professionals considered their knowledge of disaster preparedness to be high, consistent with a study conducted in Saudi Arabia, which showed a satisfactory level of knowledge among the targeted population (Mohamed, Abdel-Aziz & Elsehrawy, 2023). In contrast, low levels of preparedness were observed by Khan et al. (2021). The variations across the studies indicate that region-specific policy-driven interventions are required. This study showed that nurses had significantly higher knowledge, attitude, and practice scores compared to physicians, pharmacists, and other healthcare professionals. Nurse professionals exhibit better preparedness knowledge, possibly due to different curricula, access to professional development, and more direct patient interaction, which nurses experience more often. The workforce requires specialized training to advance all healthcare professionals, including physicians, technicians, and allied medical staff, to achieve equal competencies in emergency preparedness and response. Furthermore, the association between marital status and knowledge scores should be interpreted with caution, as it may simply reflect underlying factors such as age or work experience rather than being an independent predictor.

Regarding HCPs’ attitude towards disaster preparedness, most respondents had a positive attitude toward disaster management, with 40% expressing interest in educational classes in disaster medicine, and 43.6% believed that disaster preparedness is essential for all healthcare professionals. Previous studies revealed that participants who had prior disaster education or participated in a disaster simulation exercise showed significantly increased levels of perceived disaster preparedness (Aslanoğlu et al., 2024; Lin et al., 2024). The present study confirmed the positive influences of disaster-specific education and simulation drills on disaster knowledge and individual perceptions of preparedness. In addition, more than one-third of the participants in this study reported that training is mandatory for all healthcare professionals. A study conducted by Ogedegbe et al. (2012) depicted the importance of training on disaster preparedness among healthcare professionals. This encouraging finding suggests that healthcare professional education requires immediate integration of disaster training, which must become mandatory for all HCPs. The potentially stronger nursing professional knowledge base can enhance emergency response capabilities by implementing interprofessional mentorship or collaborative exercises (Al Thobaity, 2024; Sultan et al., 2023). A study by Corrigan & Samrasinghe (2012) to assess disaster preparedness in an Australian urban trauma center showed that 59.3% of the respondents had previously received training to handle a disaster, 37.9% had attended a disaster simulation drill, and 12.9% had actually experienced a disaster. The study further demonstrated that these participants were more prepared to deal with such situations than those with no such disaster handling or simulation experience (Corrigan & Samrasinghe, 2012).

The practice-related findings of this study highlight areas of both strength and concern within the current state of disaster readiness among HCPs in central Saudi Arabia. For example, some items, including updating readiness by reviewing the guidelines and being comfortable with performing disaster management protocols, received positive responses from nearly half of the participants. However, we found a significant lack of preparedness for other items. The preparedness level among the HCPs differs across the studies (Gillani et al., 2022; Shanableh et al., 2023). The variation in findings is attributed to the study settings and their participants. For example, regarding the practice towards disaster preparedness, 44.1% of the participants feel comfortable executing evacuation protocols in the event of a disaster, a finding reinforced by Alhamaid et al. (2024) who reported that 48.54% of the participants approved the execution of disaster drills in the hospital. Furthermore, we observed that non-Saudi nationals had significantly higher levels of preparedness than Saudi nationals. This could be attributed to the overall trend of higher readiness levels among nurses to handle disasters. Interestingly, a recent study revealed a similar result (Thirunavukkarasu et al., 2022). Another important finding identified by the present study is the positive correlation across the domains. This finding suggests that the targeted training program must be holistic to improve the HCPs’ knowledge, beliefs, and practical competencies. Interestingly, knowledge and practice did not vary significantly by work setting (primary, general, or tertiary hospitals). This suggests that the higher preparedness scores observed among nurses may be attributed to their professional training and role in patient care, rather than to differences in clinical setting or specialty exposure.

The results emphasize the current requirement for established training systems that concentrate on disaster models for enhancing readiness capabilities. Healthcare facilities need to organize regular evacuation training as well as provide specific emergency protocol development and applied staff training to boost HCPs’ disaster and emergency preparedness levels. This can be done by numerous methods, such as simulation, artificial intelligence, etc (Bari et al., 2023; Doumi et al., 2024; Sa’d & Malak, 2025). In Saudi Arabia, despite the disasters that have already occurred, there is a lack of a multi-sectoral department that aims to facilitate effective disaster health management. Instead, Saudi Arabia continues to take a traditional health approach to their response to disasters and emergencies. The ICN Framework of disaster nursing competencies recommends comprehensive disaster management plans that can be used to meet the challenges associated with disasters ranging from the lowest to the highest level.

The present study consists of several strengths. Firstly, it is one of the few studies that aimed to evaluate the HCPs of all cadres from various settings, such as primary health centers and general and specialty hospitals. Secondly, we have utilized a standard and validated data collection tool, enabling policymakers to make informed, evidence-based decisions. Furthermore, the response rate (more than 80%) among the invited participants supports the validity of our findings. Finally, we performed this study in the region where data related to HCPs are least reported in previous studies. The limitation of the present study is its cross-sectional nature, which introduces subsequent recall bias and reduce temporal robustness. The absence of triangulated qualitative data constrains the ability to explore contextual and experiential dimensions of disaster preparedness in greater depth. Therefore, while the findings provide useful baseline insights, they should be interpreted with caution when extrapolated to a larger level. In addition, there is a lack of generalizability as this study is confined to HCPs in the central region of Saudi Arabia. Furthermore, the deeper and more subjective exploration of HCPs’ preparedness cannot be stated from this study (as our study relied on self-reported survey data), as it can only be conducted through a mixed-methods or qualitative study.

Conclusions

Participants of the present study had low knowledge (36.2%), attitude (45.9%), and practice (49.2%) towards disaster preparedness. This study revealed significant positive correlations among knowledge, attitude, and practice, as well as between knowledge and attitude. Nursing professionals had significantly higher knowledge, attitude, and practice scores. The present study recommends that HCPs should receive ongoing training and skill development to handle a variety of disaster events effectively. Healthcare facilities need to organize regular disaster training programs and provide specific emergency protocols to boost HCPs’ disaster and emergency preparedness levels. Building on the observed higher knowledge levels among nurses, future preparedness efforts could explore interprofessional mentorship or collaborative training approaches to strengthen preparedness across healthcare groups. Finally, we suggest exploring the qualitative component of HCPs’ disaster preparedness using the sequential mixed-methods surveys.

Supplemental Information

Supplemental Information 1 Raw data

Supplemental Information 2 Data collection tool

Supplemental Information 3 Supplementary tables

We thank Khaled Alhuwaymili, Abdulmajeed Alshehri, Sultan Almutairi, Sultan Althobaiti, Khalid Almutairi, and Fahad Alzhrani, MPH students of Jouf University, for helping coordinate with the healthcare professionals and data collection.

Additional Information and Declarations

Competing Interests

Author Contributions

Human Ethics

Data Availability

The authors declare there are no competing interests.

Ashokkumar Thirunavukkarasu conceived and designed the experiments, performed the experiments, analyzed the data, prepared figures and/or tables, authored or reviewed drafts of the article, and approved the final draft.

Aseel Awad Alsaidan conceived and designed the experiments, authored or reviewed drafts of the article, and approved the final draft.

Sayer T. Aldhafeeri conceived and designed the experiments, authored or reviewed drafts of the article, and approved the final draft.

Doaa Mazen Abdel-Salam conceived and designed the experiments, analyzed the data, prepared figures and/or tables, authored or reviewed drafts of the article, and approved the final draft.

Nasser T. Aldhafeeri performed the experiments, authored or reviewed drafts of the article, and approved the final draft.

Muzun Ali Alanazi performed the experiments, prepared figures and/or tables, authored or reviewed drafts of the article, and approved the final draft.

Randa Mansour Almughamisi performed the experiments, prepared figures and/or tables, authored or reviewed drafts of the article, and approved the final draft.

Diyam Khalid Alkawkbei conceived and designed the experiments, prepared figures and/or tables, authored or reviewed drafts of the article, and approved the final draft.

Wijdan Al-Zahrany performed the experiments, authored or reviewed drafts of the article, and approved the final draft.

The following information was supplied relating to ethical approvals (i.e., approving body and any reference numbers):

The regional ethics committee, Gassim region, Saudi Arabia (607/46/3231) approved this study.

The following information was supplied regarding data availability:

The raw SPSS data used for analysis is available in the Supplemental File.

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
