# Peer review of "Evaluation of awareness, attitudes, and practices towards disaster preparedness among Saudi healthcare professionals: implications for enhancing emergency response and training"

_PeerJ, doi:10.7717/peerj.20464_

## Round 0.1 · original submission · Major Revisions

· Academic Editor

Major Revisions

**Language Note:** The review process has identified that the English language must be improved. PeerJ can provide language editing services - please contact us at [email protected] for pricing (be sure to provide your manuscript number and title). Alternatively, you should make your own arrangements to improve the language quality and provide details in your response letter. – PeerJ Staff

Reviewer 1 ·

Basic reporting

The manuscript is written in understandable English; however, the language requires revision to improve clarity and flow. Some grammatical and syntactical errors, awkward phrasing, and inconsistent terminology throughout the text detract from its readability and professional presentation.

The introduction provides a background on disaster preparedness but fails to critically frame the existing knowledge gaps with sufficient precision, particularly within the Saudi context. Some citations are outdated or used in a cursory manner. The authors repeatedly assert the novelty of their regional focus (Gassim), yet do not convincingly demonstrate how the present findings fill a meaningful research void beyond geography.

Figures and tables are relevant and informative, but their labeling and captions lack detail and should be improved for standalone interpretability. Notably, only 37 references are included, which is inadequate for a manuscript of this scope. Several key studies on disaster preparedness in healthcare settings are missing, particularly those on the methods part. I would recommend some changes to help this manuscript.

Experimental design

The study adopts a cross-sectional survey approach, which is appropriate for exploring awareness, attitude, and practice (KAP) levels. However, there are critical methodological concerns that should be addressed. The use of convenience sampling severely undermines the generalizability and validity of the findings. Despite acknowledging this limitation, the authors should introduce support for this method.

The KAP instrument is claimed to be standard and validated, but no evidence of published psychometric validation is provided. The authors mention Cronbach's alpha for internal consistency, but fail to provide details on construct validity or dimensionality checks. The use of Bloom’s cut-off criteria is described but not justified or referenced properly. More transparency is needed in how cut-off values were determined, especially since KAP domains are treated as ordinal outcomes in subsequent logistic regressions.

Furthermore, the study design lacks temporal robustness, as all data were collected over a short period in a single province. The authors’ assertion that this work provides actionable policy guidance is difficult to accept given the sampling limitations and absence of triangulated qualitative findings. Finally, while ethical approval and informed consent are noted, no evidence is given that the tool was culturally adapted or pre-tested for this specific context beyond brief pilot testing.

Validity of the findings

The results are presented in an orderly fashion, and the statistical analyses (Spearman correlation and logistic regression) are appropriate in general.

The reported correlations among knowledge, attitude, and practice are moderate to strong, but their causal or directional interpretations are speculative given the study design. The multivariate models lack sufficient adjustment and risk overfitting, as many predictors are entered simultaneously without justification. More transparency is needed regarding collinearity diagnostics and model selection procedures.

The conclusions drawn are overly ambitious, especially recommendations to develop interprofessional mentorship programs based solely on cross-sectional data. While such suggestions are intuitively appealing, they are not grounded in the findings and appear more as speculative extrapolations.

The authors are also encouraged to avoid repeatedly reiterating the same implications across the abstract, discussion, and conclusion.

·

Basic reporting

-

Experimental design

-

Validity of the findings

-

Additional comments

Thank you for the opportunity to review this manuscript. In this paper, the authors report on the results of a cross-sectional study of healthcare professionals in Saudi Arabia regarding their familiarity with disaster preparedness principles. The paper is well-written and clear; there are a handful of minor grammatical errors that I defer to the editors for review.

My main observation is that the survey may conflate perceived knowledge of disaster preparedness principles with actual knowledge, given that the results are based on self-reported factors. A person with a small amount of knowledge may overestimate their expertise and report a high level of familiarity with a concept, for example, while an experienced person may be aware of the limitations of their expertise and thus report a lower level of knowledge despite having a reasonable command of a topic. My main question to the authors is whether there was some objective way to measure participants’ knowledge beyond their self-reporting in the surveys?

Specific comments:
Line 57 – While urbanization and population growth have certainly contributed to disasters, the statement that these are the two principal causes of disaster is an assertion that requires at least a reference with some evidence.
Line 99 – Some discussion of the demographic characteristics of Gassim would be useful for an international readership, e.g., population, size, etc. (I just learned about Gassim’s high population density compared to other regions as well as its low poverty rate, for example.)
Line 252 – Nurses’ higher level of disaster preparedness knowledge is interesting, and I am unsure if this is universal across regions. Is there any information about the specialties of the HCPs participating in this study? I would not be surprised if ICU, emergency department, and hospital ward nurses were more familiar with disaster concepts than urologists, for example (with no disrespect intended to our urologic colleagues).
Table 1 – The association between marital status and knowledge of disaster preparedness is interesting but of uncertain value. Is this just a surrogate for age and experience (i.e., younger HCPs are less likely to be married)? I am also a little uncertain about its relevance, although I would be interested to read the authors’ thoughts about this.

---

## Round 0.2 · accepted · Accept

· Academic Editor

Accept

Dear Authors

We are pleased to inform you that your revised manuscript has undergone expert review and has been deemed to meet the standards required for publication in PeerJ. The reviewers have acknowledged the substantial improvements made during the revision process, and we commend your diligence and responsiveness to feedback.

Your submission is now formally accepted for publication. This endorsement reflects the quality and relevance of your research, as well as your valuable contribution to the scientific community.

On behalf of the editorial team, I would like to express our sincere appreciation for your efforts and commitment to advancing knowledge in your field. We look forward to receiving future submissions of your research and review articles.

Best Regards

Yung-Sheng Chen, Ph.D.
Academic Editor

Reviewer 1 ·

Basic reporting

good revisions

Experimental design

good revisions

Validity of the findings

good revisions

Additional comments

good revisions

·

Basic reporting

Thank you for the opportunity to review this revised manuscript. I have reviewed the current version, the authors' responses to my comments, and those made in reference to the other reviewer's comments. I am generally satisfied with these responses.

Experimental design

No comment - the authors note the limitations in their study design and the potential gap between perceived versus actual disaster readiness.

Validity of the findings

The findings are internally valid.

Additional comments

No additional comments.